# Effects of Jumping Phase, Leading Limb, and Arena Surface Type on Forelimb Hoof Movement

**DOI:** 10.3390/ani13132122

**Published:** 2023-06-27

**Authors:** Christina M. Rohlf, Tanya C. Garcia, Lyndsey J. Marsh, Elizabeth V. Acutt, Sarah S. le Jeune, Susan M. Stover

**Affiliations:** 1Department of Surgical and Radiological Sciences, University of California-Davis, Davis, CA 95616, USA; tcgarcia@ucdavis.edu (T.C.G.); sslejeune@ucdavis.edu (S.S.l.J.); smstover@ucdavis.edu (S.M.S.); 2Biomedical Engineering Graduate Group, University of California-Davis, Davis, CA 95616, USA; 3Animal Biology Graduate Group, University of California-Davis, Davis, CA 95616, USA; ljmarsh@ucdavis.edu; 4Clinical Large Animal Diagnostic Imaging, University of Pennsylvania, Philadelphia, PA 19104, USA; evacutt@upenn.edu

**Keywords:** equine, show jumping, hoof slide, motion capture, arena surface, leading limb

## Abstract

**Simple Summary:**

The mechanical behavior of arena surfaces has been identified as a contributor to injuries of performance horses. Evidence of excessive fetlock extension in association with stiff surfaces has propelled the installation of synthetic surfaces in performance horse arenas to reduce injury risk. However, the effect of arena surface properties on hoof slide during show jumping has not been widely studied. Therefore, this study measured the forelimb hoof motion of horses during takeoff and landing from a 1.1 m jump using a high-speed video motion capture system on five dirt and seven synthetic surfaces. Hoof slide was not significantly different between dirt and synthetic surfaces, but it was greater at takeoff than at landing and greater for the leading limb than for the trailing limb. These results indicate that horses are able to compensate for the effect of surface differences on hoof slide at a moderate jump height.

**Abstract:**

During the stance phase of equine locomotion, ground reaction forces are exerted on the hoof, leading first to rapid deceleration (“braking”) and later to acceleration (“propulsion”) as the hoof leaves the ground. Excessive hoof deceleration has been identified as a risk factor for musculoskeletal injury and may be influenced by arena surface properties. Therefore, our objective was to evaluate the effect of arena surface type (dirt, synthetic) on hoof translation of the leading and trailing forelimbs during jump takeoff and landing. Solar hoof angle, displacement, velocity, and deceleration were captured using kinematic markers and high-speed video for four horses jumping over a 1.1 m oxer at 12 different arenas (5 dirt, 7 synthetic). Surface vertical impact and horizontal shear properties were measured simultaneously. The effects of surface type (dirt, synthetic), jump phase (takeoff, landing), and limb (leading, trailing) on hoof movement were assessed using ANOVA (*p* < 0.05), while the relationships of hoof movement with surface mechanical properties were examined with correlation. Slide time (*p* = 0.032), horizontal velocity of the hoof (*p* < 0.001), and deceleration (*p* < 0.001) were greater in the leading limb, suggesting a higher risk of injury to the leading limb when braking. However, surface type and jump phase did not significantly affect deceleration during braking.

## 1. Introduction

The mechanical properties of equine arena surfaces have been proposed as extrinsic risk factors for musculoskeletal injury [1]. During the stance phase of equine locomotion, ground reaction forces are exerted on the hoof, leading to first rapid deceleration (“braking”) and later acceleration (“propulsion”) as the hoof leaves the ground [2]. When the hoof contacts the ground, deceleration occurs first in the vertical direction [3], followed by hoof slide as the center of mass of the horse moves over the planted leg, pushing that leg forward [4]. Researchers have speculated that a shorter slide duration and greater horizontal hoof deceleration may increase musculoskeletal injury risk by imposing a large bending moment on the third metacarpal bone [5]. Shorter periods of horizontal hoof braking were also associated with greater longitudinal deceleration and high-frequency oscillations of the third metacarpal bone, which may increase the risk of damage to subchondral bone and cartilage [6]. The distance and duration of hoof slide is expected to be influenced by the surface and speed of the horse [1]. In one study, hoof deceleration was greater during trot on a sandpaper surface compared to a sand surface [7], suggesting that arena surface properties may have a significant effect on hoof deceleration.

The support, or midstance, phase follows the impact and slide phases and is characterized by relatively little hoof movement and peak vertical loads. If hoof displacement during support is large, the horse may be required to apply more muscular force to maintain speed, making the horse more susceptible to fatigue [8]. Finally, the grab, or rollover, phase occurs when the hoof rotates into the surface and lifts from the ground. The hoof is able to rotate further into surfaces with lower shear strength during grab, which may reduce strain in flexor tendons and prevent slippage [9].

Desired surface characteristics of arena surfaces, as reported by riders, significantly differ by discipline [10]. Show jumping is characterized by higher ground reaction forces [11] and higher velocities at both approach and landing than other disciplines [12,13], suggesting that a jumping surface must support higher strain rates and higher vertical and shear loads [1]. Therefore, to understand the relationship of arena surface properties and musculoskeletal injury risk of equine athletes, it is important to study hoof movement on competition surfaces during sport-specific equine locomotion.

Hoof movement during locomotion has been previously recorded with accelerometers [7,14,15,16] and high-speed motion capture [17,18]. The effect of limb (leading, trailing) on hoof movement during jumping has been previously reported, where the leading limb at landing exhibited a lower hoof angle, higher horizontal hoof velocity, and greater deceleration [18]. However, Hernlund et al. did not evaluate the effect of surface type or surface properties on hoof motion during jumping.

The effect of surface type on hoof movement during a gallop has been previously characterized for the hindlimb, where dirt surfaces exhibited significantly higher horizontal motion of the heel during slide than synthetic surfaces [17]. Additionally, a higher horizontal and vertical displacement of the forelimb hoof was observed on an all-waxed sand track compared to a turf track at cantering speeds [16]. By understanding the relationship between surface type, surface composition, or surface management and hoof movement, it may be possible to modify arena surfaces to optimize hoof decelerations and improve the safety of all horses that train and compete on the surface.

Therefore, our first objective was to characterize the angle, displacement, velocity, and deceleration of the forelimb hoof during jumping to evaluate the effect of limb (leading, trailing), jump phase (takeoff, landing), and surface type (dirt, synthetic). We hypothesized that horizontal hoof deceleration would be greater for the leading limb, in line with a previous study [18]. Furthermore, surfaces with greater shear forces were expected to have greater hoof deceleration; however, since shear forces were not significantly different between dirt and synthetic surface types [19] when considering a large variety of surfaces, hoof deceleration was also not expected to be different between surface types.

Our second objective was to identify relationships between hoof movement parameters and measured properties of the arena surface (compositional, manageable, shear, and vertical impact). We hypothesized that hoof horizontal motion would be negatively correlated with shear properties (i.e., adhesion, coefficient of friction, and shear force) and vertical hoof motion would be positively correlated with the vertical displacement of the surface tester and negatively correlated with other vertical impact properties such as surface stiffness and vertical deceleration at impact.

## 2. Materials and Methods

### 2.1. Study Design

A repeated measures study design was used to track solar hoof angle, position (horizontal and vertical displacements), velocity, and acceleration for four horses (one mare, three geldings; age 9.8 ± 2.1 years; weight 544 ± 66 kg) jumping three times over a 1.1 m oxer at twelve arenas in northern California (five dirt, seven synthetic). The weight of the rider and tack used for all horses was 84 kg. Horses were visually observed for orthopedic pain and discomfort by a licensed veterinarian at the beginning of each testing day (SS). No horses showed signs of lameness during the testing period. However, one horse (gelding) was unavailable for the first testing day (dirt surface) and thus only jumped on eleven of the twelve arena surfaces. These data were collected under an IACUC protocol issued by the University of California Davis (Protocol 19843-Effect of Jumping Arena Surfaces on Equine Forelimb Biomechanics; approved 3 April 2017).

Synthetic surfaces had greater than 1% fiber content by volume; none contained rubber, oil, or wax. Fibers were primarily polyester; however, polyethylene, polypropylene, and nylon fibers were also observed in some synthetic surfaces. One dirt surface had 0.6% fiber with large felt strips poorly integrated in the cushion layer. All other dirt surfaces contained 0% fiber content.

The oxer was preceded by a ground pole and two cross rails (0.56 m from ground to center) to center and standardize the approach of the horse to the measured jump (Figure 1). Horses were accustomed to the equipment by performing 3 trot trials through the jump grid without the jump poles, followed by practice jump trials which systematically added the jump elements preceding the oxer and increased the height of the oxer. Subsequently, data were collected for 3 jump trials with the complete jump grid and final oxer height (1.1 m). The takeoff and landing zones of the oxer for each surface were harrowed using a rake between each recorded jump.

### 2.2. Measurement of Hoof Movement

Two kinematic markers were placed on a 3D-printed extension bar and rigidly attached to the hoof wall to capture solar hoof angle, translation, and velocity while the hoof was submerged in the surface material during stance (Figure 2). The extension bar consisted of a rectangular block, which was screwed into the hoof via a patch of PMMA layered with fiberglass cloth, and a hoof wand, which was attached to the hoof block with screws prior to taking measurements. Radiographs (mediolateral projection) of the hoof, which also captured the positioning of the hoof wand, were used to translate hoof wand marker positions to virtual points at the most dorsal and palmar portions of the hoof at the solar margin (“toe” and “heel”, respectively). The hoof block remained on the hoof between testing days to ensure consistent placement of the extension bar on the hoof. Two monochrome high-speed video cameras (S-PRI, AOS Technologies, Baden, Switzerland, 1280 × 1024 p, 500 fps) were centered on and calibrated in the field of view of the respective takeoff and landing zones to capture marker movement. Additionally, a wide-angle field-of-view camera (PROMON, AOS Technologies, Baden, Switzerland, 1280 × 720 p, 120 fps) and a marker on the girth were used to track true horse jump height and horizontal velocity prior to takeoff. For each jump trial, a consistent, trained observer (CR) determined whether the instrumented (left) limb was the leading or trailing forelimb at takeoff and landing. At takeoff, the instrumented forelimb was considered leading if it was the last forelimb to leave the ground; at landing, the instrumented forelimb was considered leading if it was the last forelimb to land on the ground. A second trained observer (LM) determined the video frames corresponding to hoof contact with the surface and all joint angle data were truncated to this stance phase. Marker motion was tracked throughout stance (Vicon Motus 10.0, Contemplas GMBH, Kempten, Germany).

The hoof angle of the solar surface of the hoof (SHA) as well as horizontal and vertical translation, velocity, and acceleration of the toe and heel were determined from marker motion using custom software (MATLAB, The MathWorks Inc., Natick, MA, USA). Solar hoof angle was defined relative to the arena surface where an angle of 0 degrees indicates that the solar surface of the hoof was parallel with the arena surface. Positive SHAs represent a hoof orientation with the dorsal wall of the hoof rotated counterclockwise with respect to the arena surface (toe down orientation); negative SHAs represent a hoof orientation with the dorsal wall of the hoof rotated clockwise with respect to the arena surface (toe up orientation) (Figure 3). Minimum, maximum, and average SHA during stance was determined for each jump trial.

Hoof translation data were also used to further divide the stance phase into three previously defined subphases [17] by a trained and consistent observer (CR): slide, support, and grab. Slide was defined as the period between the start of stance and the end of horizontal hoof motion, support was defined as the period between the end of horizontal hoof motion and start of vertical hoof motion, and grab was defined as the period between the start of vertical hoof motion and the end of stance (Figure 4). For each subphase of stance, horizontal displacement, vertical displacement, and average velocity for both the toe (dorsal extremity of the hoof in the sagittal plane) and heel (palmar extremity of the hoof in the sagittal plane) were recorded. Average deceleration during the slide phase (braking) is also reported. Displacement is defined as the difference in the position of the toe or heel between the start and end of each subphase. Horizontal displacement and horizontal velocity of the hoof are defined as positive when moving in the same direction as the center of mass of the horse. Vertical displacement and vertical velocity are defined as positive when moving farther into the arena surface.

### 2.3. Measurement of Arena Surface Properties

Both shear and vertical ground reaction forces were measured immediately following kinematic data collection of each horse at locations parallel to the jumping grid with surface testing equipment. Shear and vertical properties of these surfaces have been previously reported [19,20]. Shear forces and horizontal displacement were measured at 1613 Hz using a linear shear testing device with a surrogate hoof. Adhesion (a) and coefficient of friction (µ) were determined from shear tests using the Mohr–Coulomb equation (F_max_ = F_N_ × µ + a) in conjunction with maximum shear force (F_max_) measured by the device and the normal force (F_N_) applied to the device. Adhesion and coefficient of friction describe shear properties of the surface–hoof/horseshoe interface. Surfaces with either high adhesion or high coefficient of friction are expected to be associated with high grip and relatively low hoof slide [21]. Vertical forces, displacement, and acceleration were measured at 4545 Hz with a previously validated, portable, vertical impact device (VID) [22]. From these measured values, maximum vertical impact force and impulse were calculated from the VID force–time curve, stiffness and maximum vertical displacement were determined from VID force–displacement data, and maximum deceleration was determined from the VID acceleration data using custom software (MATLAB, The MathWorks Inc., Natick, MA, USA).

Cushion depth and surface temperature were also measured in conjunction with each horse, and samples (approximately 250 g) from the top two inches of the surface were collected in waterproof, airtight containers for moisture content and compositional analysis (fiber content, sand content, silt content, clay content, particle size, and fiber length) according to previously described methods [19].

### 2.4. Statistical Analysis

The central tendency (least squares mean/median) and variation (standard error/range) in all variables are described for normally or non-normally distributed data. The effect of jump phase (takeoff, landing), limb (leading, trailing), and arena surface material (dirt, synthetic) on SHA, displacement, velocity, and acceleration were all assessed in a single mixed-model ANOVA with horse included as a random effect and with jump height and horizontal velocity of the horse at takeoff included as covariates. Higher-level interactions between surface and phase and surface and lead were also analyzed. Normality of the ANOVA residuals was assessed using a Shapiro–Wilk test (W > 0.9). ANOVA on ranked data was used for variables with non-normally distributed residuals. The relationships of hoof motion measurements with surface composition, manageable factors, shear surface properties, and vertical impact properties were examined first using univariate regression (see Appendix A). For each hoof movement parameter, all related surface property variables with *p* < 0.20, as determined from the univariate regression results, were used as inputs in a multivariate stepwise regression model. Statistical significance was *p* < 0.05. SAS software (SAS Institute Inc., Cary, NC, USA) was used to perform all statistical analyses.

## 3. Results

### 3.1. Jump Characteristics

The jump height at the girth was 1.27 ± 0.07 m (mean ± standard deviation) for all horses. At takeoff, the average horizontal velocity measured at the girth increased during stance in preparation for the jump (5.32 ± 0.44 m/s before stance, 5.51 ± 0.42 m/s during stance, 5.97 ± 0.45 m/s after stance). At landing, the average horizontal velocity measured at the girth was lower than that at takeoff but also increased during stance (4.30 ± 0.48 m/s before stance, 4.39 ± 0.35 m/s during stance, 4.68 ± 0.45 m/s after stance). The resultant velocity at takeoff was directed upward, toward the jump (5.91 ± 0.43 m/s; 13.6 ± 2.8° from horizontal) and had a higher magnitude than the resultant velocity at landing, which was, on average, directed toward the ground and away from the jump (4.70 ± 0.36 m/s; −1.83 ± 2.7° from horizontal).

### 3.2. Characteristics of Hoof Motion during Stance

The contact time for the hoof was significantly longer at takeoff than at landing (0.212 ± 0.006 s takeoff, 0.202 ± 0.006 s landing, *p* < 0.001), and also significantly longer when the instrumented limb was the leading limb (0.212 ± 0.006 s leading, 0.201 ± 0.006 s trailing, *p* < 0.001). Surface type (dirt, synthetic) did not have a significant effect on contact time (*p* = 0.438). However, when evaluating combined effects, the effect of leading limb on contact time was greater on dirt surfaces than on synthetic surfaces (*p* = 0.028).

Average hoof angular motion for takeoff (gray) and landing (black) is depicted in Figure 5. At the beginning of stance, the SHA was noticeably lower (more toe-up) at takeoff than at landing. The average SHA for dirt (brown) and synthetic (gray) surfaces at takeoff and landing is also shown in Figure 6. The solar hoof angle was not noticeably different between dirt and synthetic surfaces. Shaded regions represent a 95% confidence interval for the average hoof angular motion during the stance phase.

Hoof angle was maximal (“toe down” orientation) at the end of the stance phase for all trials (100 ± 0%). Hoof angle was minimal (“toe up” orientation) near the beginning of the stance phase for takeoff (location of minimum at 1.8 ± 8.4% stance) as the hoof started “toe up” during the stance phase of takeoff in all but 2 of the 139 takeoff jump trials (98.6% of trials “toe up”). However, at landing, there was wide variation in the location of the minimum hoof angle amongst trials (38.0 ± 23.7% stance). At landing, the hoof started stance “toe down” in 103 of the 142 jump trials (72.5% of trials “toe down”), and the minimum SHA occurred during midstance rather than at hoof strike.

### 3.3. Hoof Movement during Slide

On average, the slide phase constituted the first 25.1 ± 9.2% of the stance phase. The slide phase constituted a greater percent of the stance phase on the trailing limb than on the leading limb (23.1 ± 1.1% of stance when leading, 27.0 ± 1.1% of stance when trailing, *p* = 0.004). However, surface type, jump phase, and jump lead did not have a significant effect on the duration of slide (Table 1).

During slide, hoof angle was significantly different at takeoff than landing, with a toe-up orientation at takeoff and a toe-down orientation at landing. Hoof angle during slide was also significantly different between the leading and trailing limb, with a toe-up orientation when the limb was leading and a very slight toe-down orientation when the limb was trailing.

Horizontal toe and heel displacement during slide were in the same direction as horse travel and were significantly greater at takeoff than landing and when the instrumented limb was leading. Average horizontal velocities of the toe and heel during slide were also significantly greater at takeoff and on the leading limb. Surface type (dirt, synthetic) did not have a significant effect on horizontal hoof displacement and velocity during slide.

The horizontal deceleration of both the toe and heel was significantly greater when the instrumented limb was leading. The horizontal deceleration was not significantly different between takeoff and landing or between dirt and synthetic surface types; however, there was a significant combined effect of jump phase and surface type on the horizontal deceleration at the heel, where deceleration was higher at landing than takeoff on dirt surfaces but lower at landing than takeoff on synthetic surfaces (*p* = 0.034).

The toe and heel moved deeper into the surface during slide (reported as positive numbers). The vertical displacement and average vertical velocity of the toe during slide were greater for the leading limb, at takeoff, and on synthetic surfaces, while vertical displacement and average vertical velocity of the heel during slide were greater for the trailing limb at landing. The average vertical velocity of the heel during slide was also greater on synthetic than dirt surfaces.

### 3.4. Hoof Movement during Support

On average, the support phase constituted the middle 55.0 ± 13.4% of the stance phase. The support phase was significantly longer and a greater percent of stance at takeoff than landing (56.5 ± 1.4% of stance at takeoff, 53.7 ± 0.9% of stance at landing, *p* = 0.025) and when the instrumented limb was the leading limb than the trailing limb (57.6 ± 0.9% of stance when leading, 52.6 ± 0.9% of stance when trailing, *p* < 0.001). The hoof angle during support was significantly lower (more “toe up”) for the leading limb, at takeoff, and on dirt surfaces.

The horizontal toe and heel displacement during support was opposite to the direction of forward movement of the horse’s center of mass (reported as negative numbers). The horizontal toe and heel displacement was significantly greater at landing than at takeoff (Table 2), and horizontal heel displacement was also significantly greater on the leading limb. The average horizontal velocity of the toe and heel during support was also significantly greater at landing than takeoff. Surface type (dirt, synthetic) did not have a significant effect on horizontal hoof displacement and velocity during support.

The toe and heel also moved deeper into the surface during support (reported as positive numbers). The vertical displacement and vertical velocity of the toe during support were greater for the leading limb at takeoff, while the vertical displacement and vertical velocity of the heel during support were greater for the trailing limb at landing. Surface type (dirt, synthetic) did not have a significant effect on the vertical hoof displacement and velocity during support.

### 3.5. Hoof Movement during Grab

On average, the grab phase constituted the last 20.0 ± 3.3% of the stance phase. The grab phase was significantly longer and a greater percent of stance on synthetic surfaces than dirt surfaces (20.3 ± 0.4% of stance on synthetic surfaces, 19.4 ± 0.5% of stance on dirt surfaces, *p* = 0.023). The grab phase was also a significantly greater percent of stance for the trailing limb (19.2 ± 0.5% of stance when leading, 20.5 ± 0.5% of stance when trailing, *p* = 0.004).

The horizontal toe displacement during grab was opposite to the direction of forward movement of the horse’s center of mass, while the horizontal heel displacement during grab was in the same direction as the horse. This displacement trend is consistent with the positive (“toe-down”) SHAs observed during grab. The horizontal toe displacement during grab was significantly greater (more negative) at landing (Table 3). The average horizontal velocity of the toe during grab was also significantly greater (more negative) at landing and on dirt surfaces. The horizontal heel displacement and velocity during grab were significantly greater (more positive) at landing with the trailing limb. The horizontal heel displacement was also greater (more positive) on synthetic than dirt surfaces.

The heel displaced out of the surface during grab, while the toe sometimes penetrated deeper into the surface (positive) and sometimes further out of the surface (negative) by the end of the grab phase. The toe displacement during grab was significantly related to the surface type, where the toe displaced further out of synthetic surfaces and further into dirt surfaces during grab. Additionally, the toe of the leading limb displaced into the surface and the toe on the trailing limb displaced out of the surface during grab. The heel displacement out of the surface was significantly greater for the trailing limb at landing. The average vertical velocity of the toe was significantly greater for the trailing limb and on synthetic surfaces with a direction out of the surface. The average vertical velocity of the heel was significantly greater for the trailing limb and at landing with a direction out of the surface.

When considering the combined effects of surface type and leading limb, the toe moved further backward and had a larger horizontal velocity with the leading limb than the trailing limb on synthetic surfaces, while the toe moved further backward and had a larger horizontal velocity with the trailing limb than the leading limb on dirt surfaces (*p* = 0.006 displacement; *p* = 0.004 velocity).

### 3.6. Multivariate Stepwise Regression of Surface Properties with Hoof Movement Parameters

Descriptive statistics for all surface composition variables (fiber content, sand content, silt content, clay content, average particle size, particle size deviation, average fiber length, and fiber length deviation), manageable properties (temperature, cushion depth, and moisture content), shear properties (adhesion, coefficient of friction, and normalized maximum shear force), and vertical impact properties (maximum vertical impact force, impulse, loading rate, maximum vertical displacement, soil rebound, energy dissipated during impact, stiffness, and maximum deceleration) as reported by Rohlf et al. [19,20] are depicted in Table 4.

The relationships between surface properties and hoof translation parameters were analyzed with a multivariate stepwise regression (see Appendix A). Clay content was significantly related to many hoof movement variables at both takeoff and landing. At takeoff, surfaces with greater clay content had less displacement (r = −0.33; *p* = 0.022) and velocity (r = −0.49; *p* = 0.004) of the heel into the surface during slide. At landing, surfaces with higher clay content were significantly related to less vertical displacement (r = −0.33; *p* = 0.031) and velocity (r = −0.45; *p* = 0.003) of the toe into the surface during slide. During the grab phase of landing, surfaces with higher clay content exhibited a reduced SHA (less “toe down” orientation) (r = −0.42; *p* = 0.004), reduced horizontal heel displacement (r = −0.39; *p* = 0.009) and velocity (r = −0.5; *p* < 0.001), less vertical heel displacement (r = −0.46; *p* = 0.002; out of the surface), and less vertical toe velocity (r = −0.42; *p* = 0.005; out of the surface). Soils with more variable particle size distributions were also related to reduced horizontal heel displacement (r = −0.32; *p* = 0.038) and lower vertical toe velocity (r = −0.33; *p* = 0.030; out of the surface) during the grab phase at takeoff, and less vertical displacement of the toe out of the surface (r = −0.37; *p* = 0.013) during the grab phase at landing.

The horizontal deceleration of the hoof during slide was also related to surface characteristics. At takeoff, surfaces with greater variation in soil particle size exhibited less horizontal heel deceleration (r = −0.41; *p* = 0.005), while surfaces with higher temperatures exhibited less horizontal toe deceleration (r = −0.41; *p* = 0.005). At landing, surfaces with higher adhesion increased the deceleration of both the toe and heel (r = 0.33; *p* = 0.027).

Several surface properties were only related to hoof movement variables at takeoff. Surfaces with greater soil rebound exhibited more vertical heel displacement (r = 0.39; *p* = 0.010) and velocity (r = 0.33; *p* = 0.012) during slide. Additionally, higher surface temperatures were related to greater vertical displacement (r = 0.35; *p* = 0.023) and velocity (r = 0.41; *p* = 0.007) of the toe during support, but reduced vertical velocity (r = −0.42; *p* = 0.005) of the toe during slide. Surfaces with deeper cushion layers were also related to greater vertical displacement (r = 0.44; *p* = 0.002) and velocity (r = 0.40; *p* = 0.004) of the toe during support and a lower average SHA (r = −0.35; *p* = 0.018; more “toe up”). Surfaces with higher vertical deceleration rates were related to less vertical displacement of the heel (r = −0.33; *p* = 0.027; out of the surface) and less horizontal displacement of the toe (r = −0.32; *p* = 0.041) during grab. Surfaces with greater impulse during vertical impact (factor of impact force and duration of impact) exhibited less vertical toe displacement (r = −0.40; *p* = 0.008; into the surface) during grab. Finally, synthetic surfaces with longer fibers exhibited more vertical heel displacement into the surface during the slide phase (r = 0.62; *p <* 0.001) and more vertical heel displacement out of the surface during the grab phase (r = 0.40; *p* = 0.041).

Additionally, some surface properties were only related to hoof movement variables at landing. Surfaces with higher adhesion exhibited greater horizontal toe and heel displacement (toe: r = 0.33; *p* = 0.031, heel: r = 0.32; *p* = 0.041) and velocity (toe: r = 0.36; *p* = 0.015, heel: r = 0.36; *p* = 0.020) during slide, as well as greater horizontal heel displacement opposite the forward momentum of the horse (r = 0.32; *p* = 0.030) during support and greater vertical velocity of the heel out of the surface during grab (r = 0.41; *p* = 0.005). Surfaces with greater dissipated energy at impact (factor of impact force and vertical displacement into the surface) exhibited less horizontal toe displacement and velocity (opposite the forward momentum of the horse) during support (displacement: r = −0.32; *p* = 0.042, velocity: r = −0.32; *p* = 0.041) and grab (displacement: r = −0.46; *p* = 0.002, velocity: r = −0.37; *p* = 0.014). Stiffer surfaces (r = −0.33; *p* = 0.011), surfaces with higher loading rates during vertical impact (r = −0.28; *p* = 0.042), and surfaces with less sand content (r = 0.28; *p* = 0.019) were related to less heel displacement out of the surface during grab. Finally, surfaces with larger soil particles exhibited less vertical velocity of the toe out of the surface during grab (r = −0.28; *p* = 0.044).

Fiber content, silt content, fiber length variation, moisture content, coefficient of friction, maximum normalized shear force, maximum vertical impact force, and maximum vertical displacement of the impact tester were not related to any hoof movement properties.

## 4. Discussion

Our results appear to agree well with the previous research. Although surface type, jump phase, and leading limb were all identified as factors that may affect the horizontal deceleration of the hoof during slide and injury risk [5,6], only the leading limb had a statistically significant effect on the horizontal hoof deceleration. Furthermore, while surface properties were significantly related to some hoof movement variables, expected negative relationships between shear properties and horizontal hoof movement and expected positive relationships between vertical movement of the hoof and surface tester were not observed.

The methods used to determine slide, support, and grab in this study appear to be reasonable, since the timing of these phases matches the timings presented in previous research. In this study, the slide phase constituted the first 25% of the stance phase on average, which is in line with previous reports of secondary impact occurring in the first 30% of stance [2]. Furthermore, the timing of grab, also called breakover, in this study agreed with the timing observed in previous studies (80–100% stance in present study; 85–100% stance in Thomason et al. [2]).

The leading limb had a significant effect on many hoof movement parameters during slide. As expected, average horizontal deceleration in the leading limb was significantly greater than horizontal deceleration in the trailing limb during slide, which is also aligned with previous research findings [18]. Rapid deceleration has been linked to a greater risk of subchondral bone and cartilage damage by increasing the bending moment, longitudinal deceleration, and high-frequency oscillations of the third metacarpal bone [5,6]. Therefore, our findings suggest that the leading limb may be at greater risk of musculoskeletal injury. The finding that SHA was significantly greater for the trailing limb during all three jump phases also aligns with the results of Hernlund et al. [18]. Results from the present study also show that the horizontal velocity of the hoof is significantly greater in the leading limb during slide. Greater horizontal velocity in the leading limb at jump landing has also been reported in the literature [18]; however, in the previous study, velocity was reported for the entire stance phase and was not subdivided into slide, support, and grab.

In contrast, surface type had few significant effects on hoof motion during slide, support, and grab. Although a previous study of galloping horses found a longer support duration on a synthetic surface compared to a dirt surface [17], this was not observed in the present study. Support duration was longer and comprised a greater percent of stance on the leading limb at takeoff (the last forelimb to leave the ground), possibly to provide additional stability and time at takeoff for the horse to plant the hindlimbs. Grab time was significantly greater and comprised a larger percent of stance on synthetic surfaces; however, this finding could be explained by the fact that vertical hoof displacement was also significantly greater on synthetic surfaces and the time required for the horse to remove their hoof from deeper within the surface while pushing off was greater. A previous study also found significantly greater horizontal hoof displacement on a synthetic surface during slide [17]; however, no surface type effects on horizontal hoof motion during slide were observed in the current study. However, this previous study only evaluated one surface of each type and may have underestimated the variation in mechanical behavior within a surface type category (dirt, synthetic). The evaluation of five dirt and seven synthetic surfaces in the present study likely captured more variation and demonstrated that surface type, as a categorical variable, does not sufficiently describe the hoof–surface interaction. Additionally, since horizontal hoof motion is expected to be related to shear properties of arena surfaces [1], the lack of surface type effects on horizontal deceleration during slide agrees with our hypothesis because shear properties of these dirt and synthetic surfaces were not significantly different [19].

Our analysis of SHA also suggests a secondary mechanism of injury in show jumping in addition to rapid horizontal deceleration. During initial impact and slide, the SHA was heel-down at takeoff and toe-down at landing. Researchers have suggested that injury risk may be reduced when the hoof impacts the surface in a heel-first configuration because the structure of the hoof provides more elasticity in the heel area to dampen impact loads [23]. Furthermore, in a heel-first impact, tension in the flexor tendons decreases as the hoof rotates to a flat orientation during stance, which partially counteracts the increase in flexor tendon force due to fetlock extension [24]. In contrast, toe-first impacts have been suggested as a contributing factor for navicular disease because rotation of the hoof and fetlock extension both increase the forces applied to the deep digital flexor tendon and navicular bone [24]. Since the toe impacted first at jump landing, when impact forces are maximized, this supports a possible mechanism for the high prevalence of navicular injuries observed in non-elite show-jumping horses in the United Kingdom [25].

Finally, several significant relationships between hoof movement during jumping and arena surface properties were also identified. Surface clay content had the most significant relationships with hoof movement on arena surfaces. In general, surfaces with more clay content reduced or slowed hoof movement. The effect of clay on hoof movement is similar to the effect of clay courts on tennis balls, as the high coefficient of sliding friction of clay reduces the ball speed [26]. Additionally, surfaces with greater surface temperatures, more variably sized particles, and less adhesion were associated with less deceleration of the hoof during the sliding phase. These findings suggest that surfaces with these characteristics may also reduce the risk of injury to the horse by reducing the bending moment on the third metacarpal bone [5]. However, compositional and manageable properties were not systematically adjusted within a single arena surface. Therefore, future research is needed to understand the individualized effects of changing a single surface management or compositional parameter on hoof deceleration.

Although shear properties (adhesion, coefficient of friction, and maximum shear force) were expected to be strongly and negatively correlated with horizontal displacement of the hoof, especially during the slide phase, adhesion was the only shear property correlated with hoof movement. This hypothesis was based on previous research which showed that high shear forces (traction) restricted the amount of hoof movement [27,28]. Both adhesion and coefficient of friction have been defined as important metrics to assess friction between the playing surface and footwear of human athletes [29], where higher adhesion and/or a greater coefficient of friction are expected to reduce slip. Although adhesion was related to several horizontal hoof movement variables at landing, in all cases, adhesion was positively correlated with horizontal hoof motion (e.g., surfaces with higher adhesion exhibited more horizontal hoof motion). This result was very counterintuitive considering the mechanical nature of adhesion to prevent hoof movement.

Vertical hoof displacement was also expected to be strongly and positively correlated with vertical displacement of the vertical impact tester because the vertical impact testing device was designed to simulate the interaction of the hoof and the ground [22]. However, no relationships between vertical displacement of the hoof and vertical displacement of the tester were found. In all but 4 jumping trials (96%), hoof vertical displacement exceeded the maximum displacement of the vertical tester into the surface. Furthermore, the hoof penetrated the base surface layer (a compacted mixture of sand and gravel with a much higher stiffness than the sandy, aerated cushion layer on the top of the surface) in 63.4% of trials, while the vertical impact device only penetrated the base layer during 6 of the impact trials (6.7%). Previous studies of human playing surfaces have also found few correlations between the results of material testing devices and results from experimental subjects [30].

The counterintuitive and unexpected correlation results between hoof movement and surface properties measured with the mechanical testing devices suggest that further experimentation and development of these devices are required to better represent surface–hoof interactions during a jump. While these devices are not expected to replicate the magnitude of forces during equine stance, it is important that these devices are further validated to ensure that surface properties as measured by these devices are representative of true hoof–surface interactions. The benefit of mechanical testing devices lies in their ability to create a more standardized testing environment than is possible from live animal studies, and to reduce the number of confounding variables when comparing the behavior between different surfaces. However, even with further development, mechanical testing devices will have limitations and may not be able to represent hoof movement accurately or precisely. Therefore, it may be preferable to focus on the enhancement and development of non-invasive and low-profile equipment that can measure movement and forces directly at the hoof (i.e., dynamometric horseshoes).

Another limitation of this study is related to possible vibration of the hoof extension bar at impact which is not consistent with the sagittal plane motion of the hoof. However, large vibrations would likely have been observed while reviewing video capture, and small vibrations were removed during the high-pass filtering of the motion capture data. Additionally, dirt and synthetic arena surfaces evaluated in this study were limited to the northern California region, which may not capture the variation in arena properties worldwide. However, our sample size (five dirt; seven synthetic) was larger than that of previous studies which compared only one to two surfaces of each type, and it likely captures more variation than these previous studies. Finally, compositional and manageable properties were not varied within a particular arena surface in this study. Thus, other compositional and manageable properties may be confounding factors when trying to determine the individualized effects of surface parameters on joint motion. The confounding effect of surface temperature is especially noteworthy as dirt surfaces had significantly higher temperatures than synthetic surfaces. Although multivariate stepwise regression was chosen to statistically account for possible relationships between surface properties when reporting correlations, future research studies should evaluate the individualized effects of changing a single surface management or compositional parameter on hoof motion.

## 5. Conclusions

Large surface property variation among synthetic and dirt surfaces precluded the ability for surface type, described categorically, to be a good predictor for hoof translation in the surface. However, hoof deceleration during slide was significantly dependent on the forelimb that was leading during the jump, where the leading leg had significantly greater deceleration than the trailing leg. This finding suggests that the leading leg may have a higher risk of musculoskeletal injury. To alleviate the increased risk of injury to the leading leg, it may be beneficial for trainers and riders to switch leads during training and competition to reduce the number of repetitions of high deceleration on a single forelimb.

## Figures and Tables

**Figure 1 animals-13-02122-f001:**
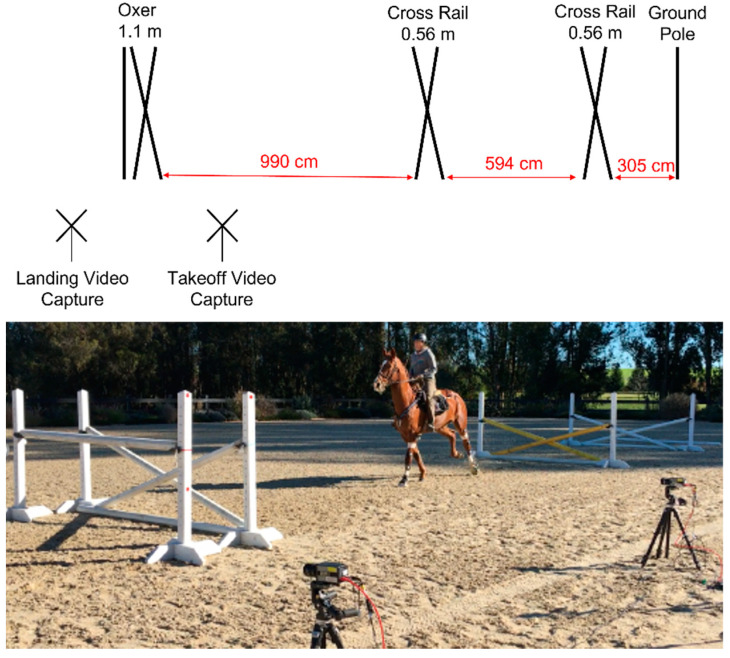
Scaled diagram of jumping grid (**top**) and photograph of jumping grid (**bottom**). Two high-speed cameras were used to record takeoff and landing of the final 1.1 m oxer.

**Figure 2 animals-13-02122-f002:**
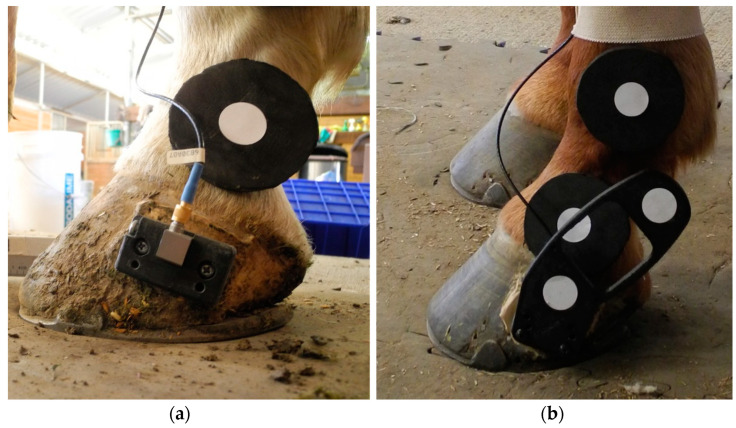
Photograph of hoof extension bar including: (**a**) A rectangular hoof block which remained rigidly attached to the hoof wall throughout the study period; (**b**) a hoof wand with kinematic markers to track hoof rotation and translation.

**Figure 3 animals-13-02122-f003:**
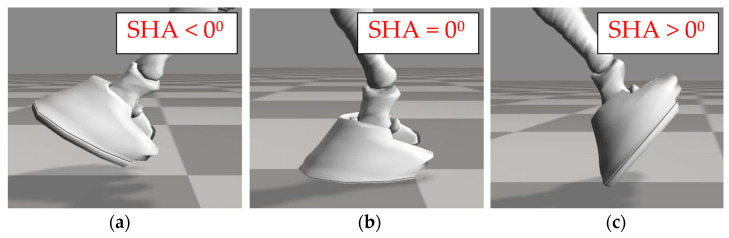
Definition of solar hoof angle (SHA): (**a**) Negative angles indicate that the dorsal wall of the hoof is rotated clockwise with respect to the arena surface (“toe up” orientation); (**b**) an angle of 0⁰ indicates that the solar surface of the hoof is parallel to the surface; (**c**) positive angles indicate that the dorsal wall of the hoof is rotated counterclockwise with respect to the arena surface (“toe down” orientation).

**Figure 4 animals-13-02122-f004:**
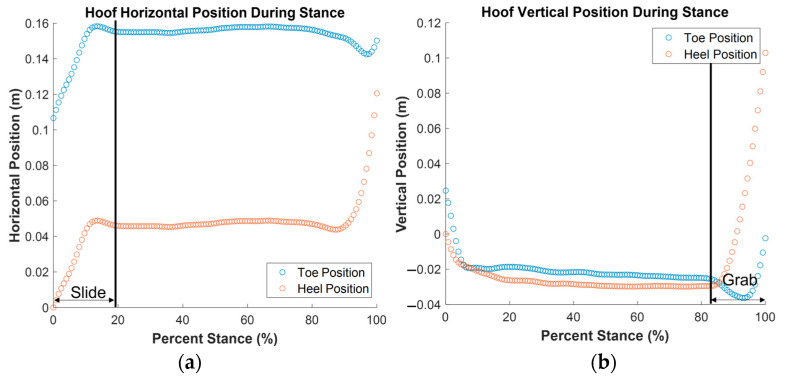
Subphases of stance: (**a**) Slide was defined as the region between the start of the stance phase and the end of horizontal hoof movement; (**b**) grab was defined as the region near the end of stance where the toe sank deeper into the surface. Support captured the period of the stance phase between slide and grab, where horizontal and vertical hoof displacement were minimal. Graphs depict hoof movement for a single horse at takeoff. Sample graphs depicting hoof movement at landing are presented in the Appendix A.

**Figure 5 animals-13-02122-f005:**
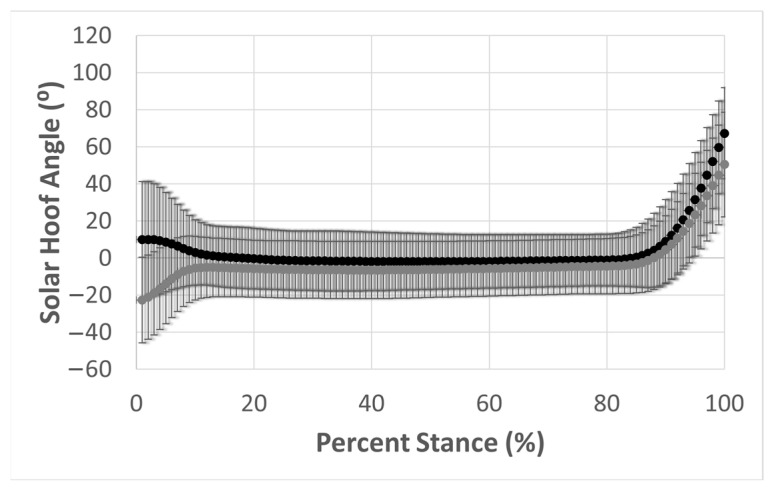
Average SHA normalized to percent stance at takeoff (gray) and landing (black). Bars indicate a 95% confidence interval for SHA at each percent stance.

**Figure 6 animals-13-02122-f006:**
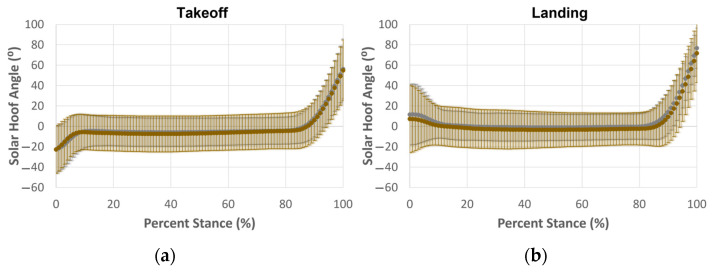
Average SHA normalized to percent stance for dirt (brown) and synthetic (gray) surfaces at (**a**) takeoff; (**b**) landing. Bars indicate a 95% confidence interval for joint angle at each percent stance.

**Table 1 animals-13-02122-t001:** The effect of jump phase (takeoff, landing), jump lead (leading, trailing), and surface type (dirt, synthetic) on hoof movement during slide (LSMeans ± SE).

Variable	Takeoff (*n* = 139)	Landing (*n* = 142)	*p*-Value	Leading (*n* = 135)	Trailing (*n* = 146)	*p*-Value	Dirt (*n* = 111)	Synthetic (*n* = 170)	*p*-Value
Slide Time (s)	0.051 ± 0.003	0.052 ± 0.003	0.527	0.049 ± 0.003	0.055 ± 0.003	0.053	0.052 ± 0.003	0.052 ± 0.003	0.709
Average Hoof Angle (°)	−8.6 ± 3.0	2.2 ± 3.0	<0.001 *	−6.7 ± 3.0	0.3 ± 3.0	<0.001 *	−3.6 ± 3.0	−2.9 ± 3.0	0.289
Horizontal Hoof Motion									
Displacement of Toe (cm)	3.6 ± 0.5	2.1 ± 0.5	<0.001 *	4.1 ± 0.5	1.6 ± 0.5	<0.001 *	3.0 ± 0.5	2.7 ± 0.5	0.108
Displacement of Heel (cm)	2.6 ± 0.5	1.6 ± 0.5	<0.001 *	3.1 ± 0.5	1.0 ± 0.5	<0.001 *	2.3 ± 0.5	1.9 ± 0.5	0.105
Average Velocity of Toe (m/s)	0.71 ± 0.11	0.50 ± 0.11	<0.001 *	0.86 ± 0.11	0.35 ± 0.11	<0.001 *	0.65 ± 0.11	0.56 ± 0.11	0.081
Average Velocity of Heel (m/s)	0.52 ± 0.11	0.42 ± 0.11	0.038 *	0.67 ± 0.11	0.26 ± 0.11	<0.001 *	0.52 ± 0.11	0.42 ± 0.11	0.075
Average Deceleration of Toe (m/s^2^)	33.4 ± 6.9	27.6 ± 6.9	0.087	46.2 ± 7.0	14.8 ± 7.0	<0.001 *	32.0 ± 7.0	28.9 ± 6.8	0.368
Average Deceleration of Heel (m/s^2^)	21.1 ± 8.1	26.3 ± 8.1	0.143	36.0 ± 8.2	11.4 ± 8.2	<0.001 *	22.6 ± 8.2	24.9 ± 8.0	0.509
Vertical Hoof Motion									
Displacement of Toe (cm)	5.3 ± 2.5	1.6 ± 2.5	<0.001 *	4.0 ± 2.6	2.9 ± 2.6	<0.001 *	3.1 ± 2.6	3.8 ± 2.5	0.002 *
Displacement of Heel (cm)	2.0 ± 2.5	3.8 ± 2.5	<0.001 *	2.2 ± 2.5	3.5 ± 2.5	<0.001 *	2.6 ± 2.5	3.2 ± 2.4	0.058
Average Velocity of Toe (m/s)	1.06 ± 0.08	0.35 ± 0.08	<0.001 *	0.81 ± 0.08	0.59 ± 0.08	<0.001 *	0.63 ± 0.08	0.77 ± 0.08	0.002 *
Average Velocity of Heel (m/s)	0.38 ± 0.05	0.72 ± 0.05	<0.001 *	0.47 ± 0.05	0.63 ± 0.05	<0.001 *	0.49 ± 0.05	0.60 ± 0.05	0.044 *

* Result is statistically significant (*p* < 0.05).

**Table 2 animals-13-02122-t002:** The effect of jump phase (takeoff, landing), jump lead (leading, trailing), and surface type (dirt, synthetic) on hoof movement during support (LSMeans ± SE; [min, max], median).

Variable	Takeoff (*n* = 139)	Landing (*n* = 142)	*p*-Value	Leading (*n* = 135)	Trailing (*n* = 146)	*p*-Value	Dirt (*n* = 111)	Synthetic (*n* = 170)	*p*-Value
Support Time (s)	0.120 ± 0.003	0.108 ± 0.003	<0.001 *	0.122 ± 0.003	0.107 ± 0.003	<0.001 *	0.115 ± 0.003	0.114 ± 0.003	0.436
Average Hoof Angle (°)	−5.9 ± 2.9	−1.9 ± 2.9	<0.001 *	−5.5 ± 2.9	−2.2 ± 2.9	<0.001 *	−4.7 ± 2.9	−3.1 ± 2.9	0.008 *
Horizontal Hoof Motion									
Displacement of Toe (cm)	Range: [−0.6, 0.9] Median: −0.03	Range: [−2.3, 0.6] Median: −0.11	<0.001 *	Range: [−0.6, 0.6] Median: −0.11	Range: [−2.3, 0.9] Median: −0.03	0.122	Range: [−2.3, 0.9] Median: −0.08	Range: [−0.9, 0.3] Median: −0.05	0.958
Displacement of Heel (cm)	Range: [−0.8, 0.9] Median: −0.06	Range: [−2.3, 0.5] Median: −0.12	0.004 *	Range: [−0.8, 0.3] Median: −0.16	Range: [−2.3, 0.9] Median: −0.03	0.015 *	Range: [−2.3, 0.9] Median: −0.13	Range: [−0.9, 0.2] Median: −0.08	0.184
Average Velocity of Toe (m/s)	Range: [−0.05, 0.08] Median: −0.003	Range: [−0.41, 0.06] Median: −0.011	<0.001 *	Range: [−0.05, 0.06] Median: −0.010	Range: [−0.41, 0.08] Median: −0.003	0.577	Range: [−0.41, 0.08] Median: −0.007	Range: [−0.09, 0.04] Median: −0.006	0.779
Average Velocity of Heel (m/s)	Range: [−0.07, 0.08] Median: −0.006	Range: [−0.41, 0.05] Median: −0.011	<0.001 *	Range: [−0.07, 0.03] Median: −0.013	Range: [−0.41, 0.08] Median: −0.004	0.168	Range: [−0.41, 0.08] Median: −0.011	Range: [−0.09, 0.02] Median: −0.008	0.148
Vertical Hoof Motion									
Displacement of Toe (cm)	Range: [−0.2, 2.9] Median: 0.32	Range: [−0.3, 1.9] Median: 0.20	<0.001 *	Range: [−0.1, 2.9] Median: 0.36	Range: [−0.3, 2.1] Median: 0.18	<0.001 *	Range: [−0.3, 2.9] Median: 0.27	Range: [−0.9, 2.2] Median: 0.08	0.515
Displacement of Heel (cm)	0.13 ± 0.04	0.22 ± 0.04	<0.001 *	0.07 ± 0.04	0.22 ± 0.04	<0.001 *	0.17 ± 0.04	0.18 ± 0.04	0.923
Average Velocity of Toe (m/s)	Range: [−0.02, 0.29] Median: 0.026	Range: [−0.02, 0.16] Median: 0.018	<0.001 *	Range: [−0.02, 0.29] Median: 0.028	Range: [−0.02, 0.20] Median: 0.015	0.002 *	Range: [−0.02, 0.29] Median: 0.023	Range: [−0.02, 0.07] Median: 0.023	0.871
Average Velocity of Heel (m/s)	0.009 ± 0.003	0.021 ± 0.003	<0.001 *	0.004 ± 0.004	0.025 ± 0.004	<0.001 *	0.015 ± 0.004	0.014 ± 0.003	0.473

* Result is statistically significant (*p* < 0.05).

**Table 3 animals-13-02122-t003:** The effect of jump phase (takeoff, landing), jump lead (leading, trailing), and surface type (dirt, synthetic) on hoof movement during grab (LSMeans ± SE).

Variable	Takeoff (*n* = 139)	Landing (*n* = 142)	*p*-Value	Leading (*n* = 135)	Trailing (*n* = 146)	*p*-Value	Dirt (*n* = 111)	Synthetic (*n* = 170)	*p*-Value
Grab Time (s)	0.042 ± 0.002	0.040 ± 0.002	0.092	0.041 ± 0.002	0.041 ± 0.002	0.940	0.040 ± 0.002	0.042 ± 0.002	0.017*
Average SHA (°)	14.7 ± 2.9	21.6 ± 3.0	<0.001 *	15.8 ± 3.0	20.5 ± 3.0	<0.001 *	17.8 ± 3.0	18.6 ± 2.9	0.304
Horizontal Hoof Motion									
Displacement of Toe (cm)	−2.0 ± 0.3	−4.1 ± 0.3	<0.001 *	−2.9 ± 0.4	−3.3 ± 0.3	0.210	−3.2 ± 0.3	−2.9 ± 0.3	0.054
Displacement of Heel (cm)	3.5 ± 0.7	4.9 ± 0.7	<0.001 *	3.6 ± 0.7	4.8 ± 0.7	<0.001 *	3.8 ± 0.7	4.7 ± 0.7	0.019 *
Average Velocity of Toe (m/s)	−0.48 ± 0.07	−0.98 ± 0.07	<0.001 *	−0.70 ± 0.07	−0.76 ± 0.07	0.419	−0.79 ± 0.07	−0.67 ± 0.07	0.026 *
Average Velocity of Heel (m/s)	1.00 ± 0.20	1.54 ± 0.20	<0.001 *	1.10 ± 0.20	1.44 ± 0.20	<0.001 *	1.19 ± 0.20	1.35 ± 0.20	0.145
Vertical Hoof Motion									
Displacement of Toe (cm)	0.1± 0.4	0.1 ± 0.4	0.830	0.3 ± 0.4	−0.1 ± 0.4	0.046 *	0.5 ± 0.4	−0.3 ± 0.4	<0.001 *
Displacement of Heel (cm)	−10.9 ± 0.4	−11.9 ± 0.4	<0.001 *	−11.0 ± 0.4	−11.9 ± 0.4	0.001 *	−11.2 ± 0.4	−11.7 ± 0.4	0.250
Average Velocity of Toe (m/s)	−0.12 ± 0.11	−0.20 ± 0.11	0.117	−0.11 ± 0.11	−0.21 ± 0.11	0.015 *	−0.07 ± 0.11	−0.25 ± 0.11	0.002 *
Average Velocity of Heel (m/s)	−2.82 ± 0.17	−3.19 ± 0.17	<0.001 *	−2.88 ± 0.17	−3.14 ± 0.17	<0.001 *	−3.03 ± 0.17	−2.98 ± 0.16	0.344

* Result is statistically significant (*p* < 0.05).

**Table 4 animals-13-02122-t004:** Descriptive statistics of compositional, manageable, shear, and vertical impact properties of dirt and synthetic surfaces (LSMeans ± SE).

Variable	Observations	Dirt	Synthetic	*p*-Value
Compositional Properties				
Fiber Content (%) ^1^	12	0.12 ± 0.98 Median 0 Range [0, 0.6]	4.03 ± 0.83 Median 3.4 Range [1.5, 10]	<0.001 ^2^
Sand Content (%)	12	84.6 ± 3.67	78.14 ± 3.11	0.210
Silt Content (%)	12	10.46 ± 2.54	12.31 ± 2.15	0.590
Clay Content (%)	12	4.83 ± 1.08	5.51± 0.91	0.639
Average Particle Size (mm)	12	0.74 ± 0.12	0.40 ± 0.10	0.051
Standard Deviation of Particle Size (mm)	12	1.00 ± 0.13	0.60 ± 0.11	0.042 ^2^
Average Fiber Length (mm) ^1^	7	N/A	27.7 ± 2.0 Median 24.8 Range [23.6, 38.3]	N/A
Standard Deviation of Fiber Length (mm) ^1^	7	N/A	11.8 ± 1.8 Median 10.7 Range [6.0, 20.9]	N/A
Manageable Properties				
Surface Temperature (°C)	59	26.0 ± 2.3	13.9 ± 1.9	0.002 ^2^
Cushion Depth (mm)	59	34.9 ± 7.5	53.2 ± 6.3	0.090
Moisture Content (%)	59	3.30 ± 2.37	9.84 ± 2.00	0.062
Shear Properties				
Adhesion (N)	46	30.0 ± 10.0	3.5 ± 8.4	0.078
Coefficient of Friction	46	0.37 ± 0.03	0.44 ± 0.02	0.065
Normalized Maximum Shear Force (F_max_/F_N_)	46	0.43 ± 0.02	0.46 ± 0.02	0.322
Vertical Impact Properties				
Maximum Vertical Impact Force (kN)	58	15.2 ± 1.3	13.7 ± 1.1	0.379
Impulse (N × s)	58	69.0 ± 1.0	72.1 ± 0.8	0.039 ^2^
Loading Rate (kN/s)	58	4679 ± 568	3155 ± 473	0.066
Maximum Vertical Displacement (cm)	58	1.76 ± 0.18	2.07 ± 0.15	0.157
Soil Rebound (cm)	58	0.11 ± 0.03	0.23 ± 0.02	0.010 ^2^
Dissipated Energy (J)	58	80.9 ± 1.5	79.8 ± 1.3	0.591
Stiffness (kN/m)	58	2477 ± 332	1602 ± 273	0.069
Maximum Deceleration (g)	58	63.6 ± 5.1	56.7 ± 4.3	0.322

^1^ Indicates that the ANOVA was performed on the ranked data because the residuals of the ANOVA were not normally distributed. ^2^ Result is statistically significant (*p* < 0.05).

## Data Availability

The data presented in this study are available on request from the corresponding author.

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
