# Peer review of "Effects of Jumping Phase, Leading Limb, and Arena Surface Type on Forelimb Hoof Movement"

_animals, 2023, doi:10.3390/ani13132122_

Round 1

Reviewer 1 Report

Brief summary

This paper provides a wealth of information, and, in my opinion, a very valuable paper to be published, as it provides detail on the properties of two different types of arena surfaces (synthetic and turf), demonstrates that the leading leg had a greater deceleration compared to the trailing leg, and importantly demonstrates that surface type may not be a good predictor of injury risk during show jumping. 

Comments

Congratulations on a well thought out paper, presenting a wealth of information. I have very few comments, most of which are just my interest in wanting to know more about the work you have done and/or clarifying of concepts I am unsure/ignorant of – so please bear with me if I ask something obvious! I’ve grouped the comments by section:

Materials and methods:

Figure 4 – Are these figures generated from takeoff or landing? I understand it’s not important for the purpose this figure is trying to illustrate – but curious how different the graphs may look. At what point is the “0” horizontal position as the heel position on graph (a) starts just after 0.08m. 

Section 2.4 Statistical Analysis Line 205 – My understanding from the supplementary multivariate analyses Table S4 is that significance was set at P>0.10, would that be right? If that is so, it is not reflected in the methods.

Results:

Formatting comment - Lots of Sections are labelled 3.2.

Section 3.2 “Hoof movement during support” Line 280 - Just a grammatical suggestion if “…were opposite to the direction of forward movement” might seem better? (Same suggestion for Section 3.2 “Hoof Movement during Grab” Line 303).

Section 3.2 “Hoof movement during support” Line 284 – Am I right in saying that the horizontal velocity of the heel during support was greater at takeoff and not at landing? Based on Table 2 that is.

Section 3.2 “Hoof movement during support” Line 291-292 – Would vertical velocity of the toe during support be greater for the leading limb at landing instead of takeoff?

Section 3.2 “Hoof movement during grab” Line 310-311 – Horizontal heel displacement greater on synthetic rather than dirt? (Based on Table 3)

Section 3.2 “Hoof movement during grab” Line 317-318 – Based on Table 3, would toe be displaced further out of synthetic and further into dirt surfaces during grab?

Multivariate Stepwise Regression of Surface Properties with Hoof Movement Parameters section – 

Table 4 – Just out of interest for me, how were the number of observations obtained? For example, you have 66 for manageable properties, 49 for shear, and 58 for vertical impact. 

Table 4 – I’m not a very confident stats person, would you please mind explaining to me why adhesion properties in Table 4 would differ so much between Dirt and Synthetic yet the p-value is >0.05?

Tables S1, S2, S3 – Just for my understanding, in these associations, were the measurements of surface properties used for the correlations analyses the same ones measured from the 12 tracks presented in Table 4? Am I right in saying then that you have not separated the analysis by surface type? So, you’ve measured a spectrum of say, temperature from the 12 tracks, independent of surface type, and related that to hoof movement? If so, were there two distinct “populations” for some of the surface properties when you tried to fit the regression models? 

Line 367-368 – Are these results based on Table S4? If so, the association, r2 and p value between displacement of the heel and vertical deceleration rates are not for the heel but for the toe, is that right? Also, I don’t seem to see an association of the vertical displacement of the toe and deceleration if this is indeed from Table S4? Have I misunderstood something?

Lines 374-375 – This question is just for my understanding: you have a result of higher adhesion exhibiting greater horizontal toe and heel displacement, which is counterintuitive to me, which you have also mentioned in the discussion, but a higher deceleration with higher adhesion as mentioned in line 376 is something that is to be expected, would that be right?

Table S4 – I wonder if a coefficient (or similar) could be included in the table so that the direction of effect of the independent variable on the surface properties might be helpful for the reader? 

Discussion

Does each horse tend to have a preferred leading or trailing limb for takeoff/landing? Or is it quite random across all the trials on the surfaces. I am interested to know what you think about how the hoof movement differences in leading or trailing limbs are specific to the horse instead of whether the limb was trailing or leading. 

Line 403 – “Thus, hoof angle matched the trajectory of the center of mass of the hose at takeoff and landing.” – Can you please clarify for me what this means?

Conclusion

This query is also for my benefit and discussion - sorry that I have not read the paper you mention by Symons et al. in Line 506 – would I be right to assume that Symons’ study was for horses galloping at faster speeds, and not for jumps but a flat racecourse study? Do you think the contrasting results you have found could be due to speed (and other similar associated factors), or do you think that it is indeed the bigger variation you observe now that you are not simply comparing one racing surface of each type which might be playing the bigger role here?  

I enjoyed this paper – thank you!

Author Response

Materials and methods:

Figure 4 – Are these figures generated from takeoff or landing? I understand it’s not important for the purpose this figure is trying to illustrate – but curious how different the graphs may look. At what point is the “0” horizontal position as the heel position on graph (a) starts just after 0.08m.

The figures that are presented here are from takeoff (now stated in the figure legend; Line 193-194). We have also included a figure for landing in the supplementary material. The most noticeable change between takeoff and landing was represented in the vertical position graph (b), where the starting y position of the toe was much higher than the heel at takeoff (toe up orientation), while the y position of the toe at landing was typically lower than the heel (toe down orientation).

For the vertical position graph, we were easily able to define a vertical position of zero at the ground surface; however, defining a zero position in the horizontal direction was much less intuitive. The zero point in the horizontal direction was defined as the location of the marker located at the girth at 0% stance. Thus, the heel position starts at 0.08m because the hoof is in front of the girth in early stance during takeoff. The distance between the heel and girth appears to be somewhat smaller at landing than at takeoff from a quick visual observation of the graphs. To avoid confusion for other readers, I have decided to regenerate (a) in this graph where the heel position starts at 0. However, our original definition of zero horizontal position did not impact the resulting data throughout the paper because we present displacement values (ie. the change in horizontal or vertical position during a stance subphase) rather than raw position values.

Section 2.4 Statistical Analysis Line 205 – My understanding from the supplementary multivariate analyses Table S4 is that significance was set at P>0.10, would that be right? If that is so, it is not reflected in the methods.

The supplementary multivariate analysis table was copied and pasted from my dissertation (where we did report all p-values less than 0.1). However, for this paper we set significance at P<0.05 (Line 235). I have now updated the supplementary table by removing p-values greater than 0.05.

Results:

Formatting comment - Lots of Sections are labelled 3.2.

Thank you. This has now been revised in the manuscript.

Section 3.2 “Hoof movement during support” Line 280 - Just a grammatical suggestion if “…were opposite to the direction of forward movement” might seem better? (Same suggestion for Section 3.2 “Hoof Movement during Grab” Line 303).

This has now been revised in the manuscript (Lines 319-320).

Section 3.2 “Hoof movement during support” Line 284 – Am I right in saying that the horizontal velocity of the heel during support was greater at takeoff and not at landing? Based on Table 2 that is.

Thank you for pointing out the discrepancy between the text and Table 2. I have verified the statistical analysis and the raw data and have discovered that two rows of the table were inadvertently transposed. The text is correct and now the table reflects the text to show that horizontal velocity of the heel was greater at landing than at takeoff.

Section 3.2 “Hoof movement during support” Line 291-292 – Would vertical velocity of the toe during support be greater for the leading limb at landing instead of takeoff?

Thank you for pointing out the discrepancy between the text and Table 2. I have verified the statistical analysis and the raw data and have discovered that two rows of the table were inadvertently transposed. The text is correct and now the table reflects the text to show that vertical velocity of the toe was greater at takeoff than at landing.

Section 3.2 “Hoof movement during grab” Line 310-311 – Horizontal heel displacement greater on synthetic rather than dirt? (Based on Table 3)

The text was in error. The sentence now reads “horizontal heel displacement was also greater (more positive) on synthetic than dirt surfaces” (Lines 349-350).

Section 3.2 “Hoof movement during grab” Line 317-318 – Based on Table 3, would toe be displaced further out of synthetic and further into dirt surfaces during grab?

The text was in error. The sentence now reads “the toe displaced further out of synthetic surfaces and further into dirt surfaces during grab” (Lines 356-358).

Multivariate Stepwise Regression of Surface Properties with Hoof Movement Parameters section – 

Table 4 – Just out of interest for me, how were the number of observations obtained? For example, you have 66 for manageable properties, 49 for shear, and 58 for vertical impact.

The number of observations was formerly not intuitive. Therefore, I have removed some of the extraneous values and have rerun the statistics with those extraneous values removed.

The reasoning for the number of observations is now as follows:

Manageable properties and vertical impact properties were measured at the beginning of the testing day and immediately after each horse completed their jump trials (n=4). On the first testing day one of the four horses was not available (now mentioned in the Materials and Methods Lines 108-110). Therefore, this accounts for 59 measurements. For one of the vertical impact tests, data was not saved properly; therefore, we lost one of the five measurements on surface two (which was a dirt surface). Thus only 58 vertical impact measurements are reported.

Shear properties were measured immediately after each horse completed their jump trials (n=4). Again, one of the four horses was unavailable on the first day of testing, so 47 shear measurements were taken. During data analysis, the shear measurements for one testing set of four on surface seven appeared unreasonable (as described in Rohlf et al. Shear ground reaction force variation among equine arena surfaces. The Veterinary Journal 2023) and were removed.  Thus only 46 shear measurements are reported.

Table 4 – I’m not a very confident stats person, would you please mind explaining to me why adhesion properties in Table 4 would differ so much between Dirt and Synthetic yet the p-value is >0.05?

If we were to perform a standard t-test comparing dirt and synthetic groups, we would expect to find a significant difference (since the difference in means is greater than two standard deviations). However, in the mixed model analysis of variance we also accounted for the 5 repeated measures on each surface and accounted for cushion depth, temperature, and moisture content as co-variates in the statistical model as reported in Rohlf et al. Shear ground reaction force variation among equine arena surfaces. We have now included the reference to our previous work in the text (Line 376). When these repeated measures and co-variates were considered in the model the difference in adhesion between dirt and synthetic surfaces was not significant.

Tables S1, S2, S3 – Just for my understanding, in these associations, were the measurements of surface properties used for the correlations analyses the same ones measured from the 12 tracks presented in Table 4? Am I right in saying then that you have not separated the analysis by surface type? So, you’ve measured a spectrum of say, temperature from the 12 tracks, independent of surface type, and related that to hoof movement? If so, were there two distinct “populations” for some of the surface properties when you tried to fit the regression models? 

Yes, the surface property measurements used to generate the data in Table 4 were the same used for the correlation. However, we only correlated the surface property measurements taken immediately after each horse with that horse. The initial measurements for manageable and vertical impact properties were not included in the correlation analysis.

Although we did not present the data separated by surface type, we did graph the data with surface type as distinct symbols. Upon doing this, we did not find distinct groupings for dirt and synthetic surfaces. Rather, dirt and synthetic surface points appeared to be located along the same linear relationship. In some cases (ie. temperature), the range of x-values for dirt surfaces was much wider than the range of values for synthetic surfaces; however, even in these cases the points still appeared to form a single linear relationship.

Line 367-368 – Are these results based on Table S4? If so, the association, r2 and p value between displacement of the heel and vertical deceleration rates are not for the heel but for the toe, is that right? Also, I don’t seem to see an association of the vertical displacement of the toe and deceleration if this is indeed from Table S4? Have I misunderstood something?

The results are based on Table S4. I have now added a sentence to the text to indicate this, “The relationships between surface properties and hoof translation parameters were analyzed with a multivariate stepwise regression (see supplementary material Table S4; Lines 384-385).

Based on Table S4 (at takeoff) there is an association between vertical heel displacement and deceleration and there is an association between horizontal toe displacement and deceleration. The text has now been corrected to state, “Surfaces with higher vertical deceleration rates were related to less vertical displacement of the heel (r=-0.33; P=0.027; out of the surface) and less horizontal displacement of the toe (r=-0.32; P=0.041) during grab” (Lines 412-414).

Lines 374-375 – This question is just for my understanding: you have a result of higher adhesion exhibiting greater horizontal toe and heel displacement, which is counterintuitive to me, which you have also mentioned in the discussion, but a higher deceleration with higher adhesion as mentioned in line 376 is something that is to be expected, would that be right?

That is correct. We would expect surfaces with higher adhesion to reduce the velocity and displacement of horizontal hoof movement, but to increase the horizontal hoof deceleration. Our findings were that adhesion was positively associated with deceleration (as expected), but also that adhesion was positively associated with hoof horizontal movement (contrary to what was expected).

Table S4 – I wonder if a coefficient (or similar) could be included in the table so that the direction of effect of the independent variable on the surface properties might be helpful for the reader? 

I have now reported r-values (with +/- directionality) in the table and the text in Section 3.6 as opposed to r^2 values.

Discussion

Does each horse tend to have a preferred leading or trailing limb for takeoff/landing? Or is it quite random across all the trials on the surfaces. I am interested to know what you think about how the hoof movement differences in leading or trailing limbs are specific to the horse instead of whether the limb was trailing or leading. 

There did appear to be some lead preference for a given horse. We attempted to vary the lead at takeoff; however, even when this was successful the horse would occasionally change lead mid jump and land on the same lead. However, we did account for the effect of horse as a random variable in our mixed model ANOVA, which may statistically account for the fact that a given horse may experience a lead preference.

Line 403 – “Thus, hoof angle matched the trajectory of the center of mass of the hose at takeoff and landing.” – Can you please clarify for me what this means?

The intention of this sentence was that at takeoff both the hoof and body of the horse slope upward, while at landing both the hoof and body of the horse are sloped downward. The relevance of this statement is questionable; however, so I have removed it to avoid confusion.

Conclusion

This query is also for my benefit and discussion - sorry that I have not read the paper you mention by Symons et al. in Line 506 – would I be right to assume that Symons’ study was for horses galloping at faster speeds, and not for jumps but a flat racecourse study? Do you think the contrasting results you have found could be due to speed (and other similar associated factors), or do you think that it is indeed the bigger variation you observe now that you are not simply comparing one racing surface of each type which might be playing the bigger role here?  

Symons study was for a flat racecourse at galloping speeds. Although the speeds at takeoff and landing during a jump are much lower, I would expect that surface trends would be consistent regardless of speed (ie. if synthetic surfaces led to more deceleration at gallop I would also expect more deceleration at the canter even though the magnitude of the deceleration is lower).

The idea about the effect of the sample size also stems from our two previous publications where we tested and compared vertical impact properties and shear properties of these same 12 surfaces. Previous studies comparing one surface of each type found that dirt surfaces had greater impact forces than synthetic surfaces or that synthetic surfaces had greater shear forces than dirt surfaces. However, when we reported our results from 5 dirt and 7 synthetic surfaces there were no statistically significant differences in vertical or shear forces. This suggests that the previous studies may have found significance only because they did not capture the variation in properties of different synthetic and different dirt surfaces.

Reviewer 2 Report

Review report

 In the introduction please add more information about hoof biomechanics, ideally with some objective measurements and standards.

Materials and methods are well described. Please add information about clinical status of the horses: were you sure that horses were free from orthopaedic pain or discomfort? Was the x-rays of the hoof performed to assess the coffin bone and DIP joint conformation? What was the body weight of the horses?

Data collection is proper designed. Where data about the realistic speed of canter and jump of individual horses collected? Were the real high of each jump collected? Oxer was 110 cm, but the question is how horses were jumped?

Results and used terminology should be improved.

Discussion is not well written. A thorough discussion of each result obtained concerning the corresponding study objective: was the tested hypothesis confirmed or not? Why? What previous evidence supports the specific result or not? It is critical to compare/contrast the result obtained with previous literature in the equine species first, then in veterinary medicine, and finally in human medicine (if not enough data are available for comparison in veterinary medicine) or in that article physical properties of each surface should be shortly described with literature references.

Conclusion must be shortened and rewritten. In that paragraph should be no references! However, the final conclusions are proper.  

Line 43: No space before [2], please correct it in whole text, there is more lack of spaces.

Line 144-146: I do not think that “toe up” and “toe down” is a correct medical term. Should be changed, e.g. distal phalanx up or P3 up. “Toe” indicated the assessment of all digit while you describe hoof biomechanics.

Line 153-154: See comment above. Word “toe” should be changed for medical/anatomical description, e.g. front hoof wall.

Line 158-160: Second part of the sentence “Average deceleration during the slide phase (braking) is also reported, since excessive deceleration during braking is predicted 159 to increase the injury risk [3].” Should be removed from methodology and maybe it is a good idea to described more about it in the introduction.

Line 216: “solar hoof angle” please use the shortcut SHA. Also applies in all text. ”Toe” should be replaced-see comments above.

Line 237-238: “meaning the hoof angle would reach its minimum at some point during mid  stance rather than at hoof strike” is a supposition/conclusion. Must be removed from results and described in discussion.

Line 230: you describe SHA and then you use just “hoof angle”. Please use the terminology described in materials and methods consistently in whole text!

Line 272: The number should be 3.3.

Line 296: The number should be 3.4.

Line 305-306: you should not explain anything in the result. The sentence must be removed.

Line 330: The number should be 3.5.

Line 398: Please add the number of reference.

Line 402-411: That the results! In my opinion this paragraph should be removed. There is lack of discussion and references to previous research.

Line 436-437: “and may contribute to a greater risk of musculoskeletal injury” – explain why?! Add more references. Your references are very poor.

Line 450-460: Please add references to the knowledge, relations and your expectations described in this paragraph

Line 461-472: Please see comment above.

Line 478-502: you started this paragraph with description of study limitation, then you compared your methodology with other studies, then you compared the surface temperature of dirt and synthetic surface. It is leading to chaos. In that paragraph you should focus only on the limitations.

Author Response

Review report

 In the introduction please add more information about hoof biomechanics, ideally with some objective measurements and standards.

Unfortunately, the equine industry has not developed a set of standards for measuring hoof biomechanics. However, we have added additional information in the introduction about the phases of stance (slide, support, and grab) and the expected motion of the hoof during each phase. We also describe what factors during each phase are beneficial and/or detrimental to injury risk and performance (Lines 45-63).

Materials and methods are well described. Please add information about clinical status of the horses: were you sure that horses were free from orthopaedic pain or discomfort? Was the x-rays of the hoof performed to assess the coffin bone and DIP joint conformation? What was the body weight of the horses?

All horses were determined to be free of orthopedic pain and discomfort and showed no signs of lameness during the study period. Each horse was evaluated at the start of each testing day by a licensed veterinarian. This information and the body weight information has now been added to the materials and methods (Lines 104-108).

Radiographs with a sagittal view (lateromedial projection) were taken on the first day that the horse participated in the study. The purpose of these radiographs was to translate the location of the marker (rigidly attached to the hoof wall) to virtual points at the most dorsal and palmar portion of the hoof (“toe” and “heel” respectively) as mentioned in Lines 135-138. Radiographs were not used to assess hoof conformation; however, all horses’ hooves were trimmed and shod by a trained farrier immediately prior to the study period.

Data collection is proper designed. Where data about the realistic speed of canter and jump of individual horses collected? Were the real high of each jump collected? Oxer was 110 cm, but the question is how horses were jumped?

A marker on the girth was used to track the real jump height of each horse and the horizontal velocity of the horse prior to takeoff (Lines 142-144). Jump height and velocity were included in the mixed model ANOVA as covariates to account for their effects in the statistical analysis. This is stated in the manuscript (Lines 201-202). We have also now included a brief overview of jump characteristics including jump height and velocity in the results (Lines 239-247).

Results and used terminology should be improved.

Solar hoof angle (or SHA) is now used consistently throughout the manuscript. The toe and heel are now defined anatomically (Lines 181-182) in the materials and methods. Solar hoof angle is also now defined anatomically (Lines 161-165). Certain sentences in the results have also been reworded as statements of fact (Lines 275-277 and Lines 344-345) to avoid confusion between results and conclusions.

Discussion is not well written. A thorough discussion of each result obtained concerning the corresponding study objective: was the tested hypothesis confirmed or not? Why? What previous evidence supports the specific result or not? It is critical to compare/contrast the result obtained with previous literature in the equine species first, then in veterinary medicine, and finally in human medicine (if not enough data are available for comparison in veterinary medicine) or in that article physical properties of each surface should be shortly described with literature references.

We have revised the discussion by adding additional references to support the findings. These additional references stem from both equine (Lines 487-499; Lines 516-517) and human movement research (Lines 518-520; Lines 534-536). We have also reorganized the limitations paragraph to enhance reader understanding (Lines 551-567). Finally, we have added an introductory paragraph (Lines 439-445) to the discussion and have structured the discussion paragraphs to elaborate on topics in the order presented in the introductory paragraph.

Conclusion must be shortened and rewritten. In that paragraph should be no references! However, the final conclusions are proper.

The reference has now been removed from the conclusion and the conclusion has been significantly shortened to one paragraph (Lines 569-576).

Line 43: No space before [2], please correct it in whole text, there is more lack of spaces.

This has been corrected throughout the text.

Line 144-146: I do not think that “toe up” and “toe down” is a correct medical term. Should be changed, e.g. distal phalanx up or P3 up. “Toe” indicated the assessment of all digit while you describe hoof biomechanics.

The toe and heel are considered functional anatomic terms that refer explicitly to the hoof itself (Robert C. McClure, Functional Anatomy of the Horse Foot, University of Missouri College of Veterinary Medicine). However, we have now provided an additional definition of the terms “toe up” and “toe down” by defining the clockwise or counterclockwise rotation of the dorsal wall of the hoof relative to the arena surface (Lines 161-165). Since we have now defined these terms we continue to use “toe up” and “toe down” in the remainder of the text.

Line 153-154: See comment above. Word “toe” should be changed for medical/anatomical description, e.g. front hoof wall.

The toe and heel are considered functional anatomic terms that refer explicitly to the hoof itself (Robert C. McClure, Functional Anatomy of the Horse Foot, University of Missouri College of Veterinary Medicine). We have now also defined the toe and heel as the dorsal and palmar extremity of the hoof respectively for those unfamiliar (Lines 181-182). Since we have now defined these terms we continue to use “toe” and “heel” in the remainder of the text.

Line 158-160: Second part of the sentence “Average deceleration during the slide phase (braking) is also reported, since excessive deceleration during braking is predicted 159 to increase the injury risk [3].” Should be removed from methodology and maybe it is a good idea to described more about it in the introduction.

The first part of this sentence (“Average deceleration during the slide phase (braking) is also reported”) has now been moved up earlier in the paragraph (Lines 182-183). We already described one mechanism for increased injury risk in the introduction, but have now also added another sentence to the introduction with a second mechanism through which horizontal deceleration could increase injury risk (Lines 48-53).

Line 216: “solar hoof angle” please use the shortcut SHA. Also applies in all text. ”Toe” should be replaced-see comments above.

The acronym SHA has been used throughout the text, except where solar hoof angle starts a sentence. Since “toe” and “heel” have now been defined anatomically (Lines 181-182) in the materials and methods we have not replaced these terms throughout the text.

Line 237-238: “meaning the hoof angle would reach its minimum at some point during mid  stance rather than at hoof strike” is a supposition/conclusion. Must be removed from results and described in discussion.

This sentence has now been reworded as a result. The sentence now states, “ At landing, the hoof started stance toe down in 103 of the 142 jump trials, and minimum SHA occurred during midstance rather than at hoof strike” (Lines 275-277).

Line 230: you describe SHA and then you use just “hoof angle”. Please use the terminology described in materials and methods consistently in whole text!

All instances of hoof angle have been replaced with solar hoof angle (or SHA) throughout the text.

Line 272: The number should be 3.3.

This has now been corrected in the text.

Line 296: The number should be 3.4.

This has now been corrected in the text.

Line 305-306: you should not explain anything in the result. The sentence must be removed.

This sentence has now been reworded as a result. The sentence now states, “This displacement trend is consistent with the positive (“toe-down”) SHAs observed during grab” (Lines 344-345).

Line 330: The number should be 3.5.

This has now been corrected in the text.

Line 398: Please add the number of reference.

This has now been corrected in the text (Line 452).

Line 402-411: That the results! In my opinion this paragraph should be removed. There is lack of discussion and references to previous research.

We have now shortened the discussion of solar hoof angle to focus on hoof angle during the initial impact phase because that is most closely tied to one of the frequently observed injuries in show jumping horses (navicular syndrome). This paragraph now suggests a secondary mechanism for injury in show jumping horses besides rapid horizontal deceleration (Lines 487-499).

Line 436-437: “and may contribute to a greater risk of musculoskeletal injury” – explain why?! Add more references. Your references are very poor.

We now revisit the references in the introduction to explain factors related to rapid deceleration that may lead to subchrondal bone and cartilage damage (Lines 456-458).

Line 450-460: Please add references to the knowledge, relations and your expectations described in this paragraph

We have now provided additional context for our hypothesis based on other equine research (Lines 516-517). We have also given motivation for why adhesion and coefficient of friction of the surfaces were measured by noting that these are comments metric to assess the foot-surface interaction for human sports (Lines 518-520).

Line 461-472: Please see comment above.

We have now provided additional context for our hypothesis based on the intention of the development of the mechanical testing device (Lines 526-527). We have again compared this finding with findings comparing material testing devices and experimental subject studies for human movement (Lines 534-536).

Line 478-502: you started this paragraph with description of study limitation, then you compared your methodology with other studies, then you compared the surface temperature of dirt and synthetic surface. It is leading to chaos. In that paragraph you should focus only on the limitations.

We address three limitations in this paragraph. The first limitation is related to marker vibration (Lines 551-554). The second limitation is related to the fact that we studied only a sample of dirt and synthetic surface and may still not be capturing the full extent of variation present in these surface types (Lines 554-559). The third limitation relates to the limitation of confounding variables when attempting to determine the effects of individualized surface property parameters on joint motion (Lines 559-567). We have now restructured this paragraph to enhance clarity for the reader. For each limitation we provide additional discussion for how we accounted for the limitation in the study design to reduce the effects of the limitation and/or possible recommendations for future studies to evaluate the effects of the limitation.

Round 2

Reviewer 1 Report

Thank you so much for your responses to my queries - I was so interested to hear back from you with some of my questions. I look forward to seeing the published version!

Just two tiny potential corrections for Supplementary Table S4:

(1) A grammatical suggestion for the title – “For support, positive values indicate more displacement or velocity opposite to the direction of horse movement of into the surface”.

(2) Now that r-values instead of r2 values are reported, should this part in the title “Partial r2 values for each variable…” be edited?

Author Response

Just two tiny potential corrections for Supplementary Table S4:

  • A grammatical suggestion for the title – “For support, positive values indicate more displacement or velocity opposite to the direction of horse movement of into the surface”.
  • Now that r-values instead of r2values are reported, should this part in the title “Partial r2 values for each variable…” be edited?

Thank you for reviewing this paper. We have now revised the table description for supplementary table S4. We now indicate that partial r-values for each variable are reported (rather than partial r2 values). We have also incorporated the phrase “opposite to the direction of horse movement” to define positive values for support and for the toe during grab to be more grammatically correct.

Reviewer 2 Report

Manuscript is well improved. The authors addressed all comments and recommendations. In my opinion the article is suitable for publication in Animals. 

Author Response

Manuscript is well improved. The authors addressed all comments and recommendations. In my opinion the article is suitable for publication in Animals. 

Thank you for reviewing this paper. Based on these comments we have not made any further changes to the manuscript.